# Phase I–IV Drug Trials on Hepatocellular Carcinoma in Asian Populations: A Systematic Review of Ten Years of Studies

**DOI:** 10.3390/ijms25179286

**Published:** 2024-08-27

**Authors:** Alok Raghav, Goo Bo Jeong

**Affiliations:** Department of Anatomy and Cell Biology, College of Medicine, Gachon University, 155 Getbeol-ro, Yeonsu-gu, Incheon 21999, Republic of Korea; alokalig@gachon.ac.kr

**Keywords:** atezolizumab, bevacizumab, hepatocellular carcinoma, sorafenib, nivolumab

## Abstract

Despite advances in the treatment of hepatocellular carcinoma (HCC) over the last few decades, treatment opportunities for patients with HCC remain limited. HCC is the most common form of liver cancer, accounting for approximately 90% of all cases worldwide. Moreover, apart from the current pharmacological interventions, hepatic resection and liver transplantation are the mainstay curative approaches for patients with HCC. This systematic review included phase I, II, III, and IV clinical trials (CTs) and randomized controlled trials (RCTs) on current treatments for patients with HCC in Asian populations (2013–2023). A total of 427 articles were screened, and 184 non-duplicate publications were identified. After screening the titles and abstracts, 96 publications were excluded, and another 28 were excluded after full-text screening. The remaining 60 eligible RCTs/CTs were finally included. A total of 60 clinical trials fulfilled our inclusion criteria with 36 drugs used as monotherapy or combination therapy for HCC. Most studies used sorafenib alone or in combination with any of the treatment regimens. Lenvatinib or atezolizumab with bevacizumab was used for HCC after initial sorafenib treatment. Eighteen studies compared the efficacy of sorafenib with that of other drugs, including lenvatinib, cabozantinib, tepotinib, tigatuzumab, linifanib, erlotinib, resminostat, brivanib, tislelizumab, selumetinib, and refametinib. This study provides comprehensive insights into effective treatment interventions for HCC in Asian populations. The overall assessment indicates that sorafenib, used alone or in combination with atezolizumab and bevacizumab, has been the first treatment choice in the past decade to achieve better outcomes in patients with HCC in Asian populations.

## 1. Introduction

The increasing incidence of hepatocellular carcinoma (HCC) poses a global health challenge [1,2]. According to a recent report by GLOBOCAN 2020, Mongolia has the highest age-standardized rate for both mortality and incidence of HCC. It is also estimated that in Asia, China alone accounts for 62.4% of the cases, followed by Japan (7.0%), India (5.3%), Thailand (4.2%), and Vietnam (4%) [3]. In Asia, liver cancer is the fifth most common cancer after thyroid, stomach, colon, and lung cancers, and it is the second most common cause of malignancy-related deaths in Asia [4]. In Asia, HCC accounts for the highest incidence and mortality among patients with liver cancer [4]. 

Over the last three decades, the annual crude mortality rate of HCC has increased in Asia. In addition to surgical intervention, several systemic therapies, including chemotherapy, immunotherapy, and molecular target-based therapies, have been proposed for advanced HCC. With technological advancements in research, molecular-targeted therapies are the mainstream approach for treating patients with HCC either alone or in combination with other drugs, especially in Asian populations. 

The etiology of HCC varies according to geographical region, as reported by a recently published study [5]. In the Asia–Pacific region, hepatitis virus infection is among the major causes of HCC; 70% of the patients from these regions have chronic hepatitis B virus (HBV) infection, whereas 20% have hepatitis C viral (HCV) infection [5]. A study from the Asia–Pacific region has reported that 75% of the patients with HCC in Japan have HCV infection [6]. 

The incidence of liver cancer varies among Asian populations. According to statistics from a recently published study, East Asian regions, including China, South Korea, and Japan, and Southeast Asian regions, including the Philippines, demonstrated a sharp decline in the incidence rate of liver cancer [7]. The same study observed a decline in the annual average percent change in the incidence rate of liver cancer in countries, including China (−1.6%), South Korea (−2.2%), and the Philippines (−1.7%), since 1978 [7]. However, a significant increase in the incidence of liver cancer has been reported in southwestern Asian countries, especially Israel [7]. HCC accounts for majority of liver cancer cases and affects 27% of the population in Thailand alone [7]. In recent decades, the incidence of liver cancer has significantly increased in Iran, Afghanistan, Qatar, Iraq, Azerbaijan, and Nepal [3].

Among Asian countries, liver cancer in South Korea is the fourth most common cancer in men and the sixth most common in women. The decrease in the incidence of liver cancer in South Korea is mainly because of the sharp decline in HBV, which is considered a major cause of HCC. Moreover, large-scale HBV vaccination has affected the incidence of HCC in the South Korean population. Despite several pharmaceutical and technological advancements, the advanced stage of HCC at the time of diagnosis in South Korea still requires serious attention. In a previous study, the 5-year survival rate of HCC among Korean patients was relatively lower than that of other cancer types owing to several effective surveillance drives among the high-risk population in South Korea [8]. 

Sorafenib is among the first Food and Drug Administration (FDA)-approved interventions that are accepted worldwide for the treatment of advanced-stage HCC. It exhibits a molecularly targeted therapeutic approach by targeting and inhibiting several pathways, including vascular endothelial growth factor receptor-2 (VEGFR-2), platelet-derived growth factor receptor (PDGFR), and extracellular signal-regulated kinase (ERK)/mitogen-activated protein kinase–ERK (MEK)/rapidly accelerated fibrosarcoma (RAF), thereby offering antiproliferative, antiangiogenic, and antiapoptotic effects [9,10]. In the Asia–Pacific phase III clinical trial (CT), sorafenib alone demonstrated a median overall survival of 6.5 months compared to placebo in patients with HCC, and, thereafter, sorafenib was approved as a first-line therapeutic approach in these patients [11]. 

Another drug known for treating HCC is regorafenib, a multikinase inhibitor that inhibits angiogenesis and oncogenesis, thereby altering the tumor microenvironment. One phase III RESORCE trial has demonstrated regorafenib as a second-line drug for HCC treatment after sorafenib treatment [12]. Similarly, another multikinase inhibitor, lenvatinib, is considered the first-line therapy for patients with unresectable HCC [13]. Sorafenib is among the first-line therapies for advanced-stage HCC in Asia, whereas atezolizumab and bevacizumab are among the second-line therapies for progressive HCC.

Moreover, owing to the high incidence and prevalence of HCC in Asia and the Asia–Pacific region, an extensive approach to the selection of appropriate therapies against HCC is necessary. Currently, the available treatment options are limited in Asia and the Asia–Pacific region; therefore, a reliable first-line therapy, without any side effects, should be selected to treat HCC. Therefore, this study aimed to distinguish between drug therapies among the approaches available for the treatment of HCC in Asian populations.

## 2. Material and Methods

### Search Strategy and Selection Criteria

This systematic review was conducted in accordance with the Preferred Reporting Items for Systematic Reviews and Meta-Analysis criteria (Figure 1). A systematic search for eligible studies in the EMBASE, MEDLINE (via PubMed), and CENTRAL (via the Cochrane Library) databases was conducted from 2013 to 2023 (Figure 2). A total of 427 articles were screened, and among them, 184 non-duplicate publications were identified. We excluded 96 publications after screening titles and abstracts and another 28 published papers after full-text screening. Finally, the remaining 60 eligible randomized controlled trials (RCTs)/CTs were included in this systematic review.

The inclusion criteria for this systematic review were CTs (phases I, II, III, and IV) and RCTs conducted on adult patients (≥18 years), including men and women with all stages of HCC, who received the intervention compared to those who received either placebo or active comparator in Asia or any multicentric trial wherein one study center was located in Asia. 

The quality of this systematic review was assessed using the grade system. Briefly, the grade system was divided into four levels: very low, low, moderate, and high. All eligible studies included in this systematic review were screened for imprecision, inconsistency, risk of bias, and publication bias. The validity and authenticity of the included studies were assessed by two independent reviewers using kappa statistics with inter- and intrarater agreements. The outcomes of the extracted studies were noted in the form of majority of the use of particular drugs for the treatment of HCC in Asia.

## 3. Results

We performed a systematic review of phases I, II, III, and IV CTs and RCTs on current treatments for patients with HCC (2013–2023). A total of 427 articles were screened, and among them, 184 non-duplicate publications were identified. We excluded 96 publications after screening the titles and abstracts and excluded another 28 published papers after full-text screening. The remaining 60 eligible RCTs/CTs were included in this systematic review (Figure 1).

A total of 60 CTs fulfilled our inclusion criteria, with 36 drugs screened for monotherapy or combination therapy for HCC. Most studies used sorafenib alone or in combination with any of the treatment regimens. Lenvatinib or atezolizumab with bevacizumab was used for HCC after initial sorafenib treatment. Eighteen studies compared the efficacy of sorafenib with that of other drugs, including lenvatinib, cabozantinib, tepotinib, tigatuzumab, linifanib, erlotinib, resminostat, brivanib, tislelizumab, selumetinib, and refametinib (Table 1). Three studies reported on the use of a combination of lenvatinib and sorafenib (Table 1). Another three studies reported on the use of nivolumab monotherapy for the pharmacological intervention of HCC, while one study utilized a combination of ipilimumab and sorafenib (Table 1) [14]. Single-arm studies reported on the use of cabozantinib, sorafenib, and immunotherapy using cytokines and enzalutamide (Table 1). Two studies reported on the treatment of HCC using ramucirumab and pembrolizumab (Table 1). This study provides comprehensive insights into effective treatment interventions for HCC in Asian populations. The overall assessment suggests that sorafenib, used alone or in combination with atezolizumab and bevacizumab, has remained the first treatment choice in the past decade for providing better outcomes in patients with HCC in Asian populations. A systematic review of the published articles found consistency in validity appraisal among the two raters, as assessed by a kappa statistic of 0.86. The weighted bar plots of the distribution of the risk of bias judgments within each bias domain are presented in Figure 3. A network visualization of the selected articles is shown in Figure 4. Altogether, these findings suggest that sorafenib, as part of a combination approach with other drugs, is the first-line treatment for patients with HCC in Asian populations. 

## 4. Discussion 

This review evaluated the drugs used to treat HCC in Asia over the past decade. Sorafenib is a multikinase kinase inhibitor with a molecular weight of 637 g/mol that inhibits protein pathways acting as anticancer agents. Sorafenib acts on RAF, vascular endothelial growth factor (VEGF), and platelet-derived growth factors receptors (PDGFR), as previously demonstrated [72]. RAF is a serine/threonine kinase that initiates the activation of gene transcription responsible for tumor promotion upon activation by the ras protein present on the membrane. Moreover, VEGF is responsible for angiogenesis in both normal and cancerous tissues, which is mediated through endothelial cell division and migration. The interaction of VEGF with VEGFRs 1, 2, and 3 promotes autophosphorylation of tyrosine receptor kinase, resulting in the activation of a cascade of downstream proteins. 

Additionally, sorafenib inhibits the activities of VEGFR-2/3, PDGFR-β, Flt3, and c-Kit [73,74]. The precise molecular mechanism underlying the antitumor activity of sorafenib remains unclear, although previously published studies have suggested that sorafenib acts on RAF/MEK/ERK-dependent or -independent protein kinases [75,76,77]. Another study demonstrated that sorafenib inhibits the expression of the β-catenin oncoprotein in HepG2 cells and activates the c-Jun N-terminal kinase (JNK) and p38MAPK pathways [78]. A similar study also observed that sorafenib is actively involved in the downregulation of several DNA repair and recombination genes (*XRCC-2, XRCC-5, FANCA,* and *FANCD2*), along with genes involved in cell cycle regulation (*CDC45L, CDC6,* and *CDCA5*) that further exert anticancer activities [78]. 

Sorafenib is associated with common adverse effects, including diarrhea and weight loss, as well as other secondary effects, such as alopecia, anorexia, and voice changes. A previously published study revealed that sorafenib has a significant survival benefit in patients with advanced HCC, although many patients demonstrated disease progression after a reduction in dosage or treatment discontinuation [11,79]. In the Study of Heart and Renal Protection (SHARP) trial, sorafenib exerted primary and acquired resistance, which hampered the survival benefit [80]. Previous studies demonstrated the antitumor activity of sorafenib monotherapy with some limitations, such as drug resistance and adverse effects, discouraging its use as monotherapy. A combination with nivolumab can resolve the problems associated with sorafenib monotherapy. Our results also demonstrated a trend toward the increased use of sorafenib combination therapy.

Nivolumab is a human recombinant monoclonal G4 immunoglobulin with anticancer activity mediated through programmed cell death receptor-1 (PD-1). T-cell response is commonly mediated through the PD-1 mechanism. The blockade of PD-1 receptors present on T-cells inhibits the proliferation of T-cells through a programmed cell death mechanism. In a recently published study, nivolumab was associated with some grade 1–2 adverse events, including the development of colitis and pneumonitis, along with increased alanine aminotransferase and aspartate aminotransferase activities [81]. 

Another anticancer drug, atezolizumab, acts by targeting PD-L1 on tumor cells, thereby preventing the binding of PD-L1 to its receptors, PD-1 and B7-1. The binding of PD-L1 to its receptor PD-1 inhibits the proliferation of T-cells, along with the inhibition of cytokine production and cytolytic activity, which in turn leads to T-cell inactivation. Similarly, T-cells and antigen-presenting cells (APCs) inhibit immune responses, including T-cell activation and cytokine release, owing to the active binding of PD-L1 to B7-1 present on T-cells and APCs [82,83]. Similar to other FDA-approved PD-1/PD-L1-targeted therapies, atezolizumab is also associated with adverse immune responses, including grade 1–4 immune-mediated colitis, hepatitis, and pneumonitis [84].

Bevacizumab is a recombinant humanized monoclonal immunoglobulin G that binds to the VEGF protein and prevents it from binding to its receptor, thereby exerting a neutralizing effect [85]. HCC is an extensively vascularized solid tumor with immense dense microvessels owing to angiogenesis. Hence, targeting VEGF is a crucial step in preventing tumor angiogenesis. Adverse reactions associated with bevacizumab include hypertension, fatigue, and proteinuria [85]. Bevacizumab can be used in combination with sorafenib to overcome these side effects.

A previously published study reported portal vein tumor invasion in 30% of Korean patients with HCC [86]. A single-center Korean RCT reported that conventional transarterial chemoembolization (cTACE) with radiation therapy had better outcomes than sorafenib monotherapy in HCC patients with portal vein invasion. However, two other RCTs conducted in the Korean population revealed that sorafenib monotherapy did not result in survival gain compared to transarterial radioembolization (TARE) [87,88]. The study concluded that TARE, sorafenib, and cTACE did not result in any survival gains [89]. 

Despite several drugs being present in the pharmaceutical market, HCC is a highly uncontrollable cancer with a tendency to metastasize to distant organs, including the lungs and stomach. Moreover, the gap between the etiology and genetic mutations contributes to poor treatment outcomes. The current boom in nanotechnology can provide new hope for the early intervention and treatment of HCC without any associated side effects, as in the case of drugs. Nanotechnology offers alternatives to several nanoparticles that have been widely employed in biomedical research related to cancer therapeutics. Nanoparticles improve the accessibility of drugs to human cells and increase their metabolic tendency along with delayed and prolonged therapeutic actions. Their modified surface area offer greater drug loading and mitigate the side effects of drugs. Their enhanced penetration and retention mechanisms, along with active targeting, provide highly specific targeted anticancer therapeutics. Owing to their low or negligible toxicity, enhanced biocompatibility, and biodegradability, anticancer nanoparticles have been the focus of research. In addition to the aforementioned characteristics, these nanoparticles also exhibit anti-inflammatory, antioxidant, and antiangiogenic effects, making them useful as anticancer therapeutics. 

The global pharmaceutical companies are steadily manufacturing new and novel molecules in the form of drugs for treatment of hepatocellular carcinoma during past decades. These industries considered Asia–Pacific, North America and Europe as the leading areas for their drug trials [90]. There are several pharmaceutical industries sponsor sites in these regions including Sun Yat-sen University, National Cancer Institute US, FUDAN University and Eastern Hepatobiliary Surgery Hospital. These drug trials for hepatocellular carcinoma (Phase 0 to Phase IV) were sponsored by the company itself in collaboration with the governments, individuals or institutions (Table 2; Figure 5). There are numerous drugs available in the market of USA and Europe to treat hepatocellular carcinoma, which includes pembrolizumab (Keytruda), nivolumab (Opdivo, Opdyta) and bevacizumab (Avastin) [90]. Pembrolizumab (Keytruda) is marketed for the treatment of hepatocellular carcinoma in USA and Europe and is an antineoplastic immunomodulating molecule that antagonist mechanism on Programmed Cell Death Protein 1 (PD1 or CD279 or PDCD1). It was first commercially approved in the year 2014 and launched in the markets of the US, the UK, Australia, France, and Germany by Merck & Co. Inc. and its subsidiaries (Rahway, NJ, USA). Another drug named nivolumab (Opdivo, Opdyta) performs antagonist action on Programmed Cell Death Protein 1 (PD1 or CD279 or PDCD1) and is a human IgG4 anti PD-1 monoclonal antibody that treats hepatocellular carcinoma. This drug was first approved in the year 2014 and launched in the market of the US, the UK, Australia, France and Germany by Bristol Myers Squibb Co and its subsidiaries [90]. According to global data, 26.80% of the clinical trials are Phase II, 20.98% are Phase III, 19.02% Phase I/II, 17.43% Phase I, while Phase II/III and Phase 0 comprised 2.38 and 2.90%, respectively (Table 2).

### 4.1. Pharmacogenetics of Hepatocellular Carcinoma in Asian Population

The Asian population presents diverse pharmacogenetic differences that influence the efficacy of several hepatocellular carcinoma drugs and the adverse drug reactions (ADRs) related to their racial/ethnic backgrounds [91,92,93]. Previous studies have reported that variants with higher frequencies are more common in Asian populations compared to other population types [80,94,95]. The Clinical Pharmacogenetics Implementation Consortium (CPIC) and the US Food and Drug Administration (FDA) have developed guidelines for adjusting treatments based on genetic variations within populations. For example, carbamazepine and clopidogrel have been shown to present ADRs in Asian populations compared to others [96,97]. Impaired gene expression, presentation of spliced variants, gene polymorphism, and mutations are among the important factors associated with poor prognosis and altered drug metabolism in Asian populations in the treatment of liver cancer, which is the fourth most deadly cancer (Figure 6). These factors severely affect the function of genes and in turn lead to a reduction of active drug molecules within intracellular tumor microenvironment along with phenotypic transitions and hampered survival pathways. *SLCO*, *SLC22A* and *SLC31A* are the gene families responsible for the transport of anticancer drugs against HCC. Mutations in these genes could affect the drug-mediated response against HCC. Other *SCLO* gene families including OATP1B1 (*SLCO1B1*) and OATP1B3 (*SLCO1B3*) transport sorafenib and possess redundant substrate specificity [98]. Past studies reported that single nucleotide polymorphisms (SNPs) of OATP1B1 and OATP1B3 severely affect the pharmacokinetics of statins and paclitaxel [99]. Authors of another study reported germline mutations in OATP1B1, c.388A>G (p.Asn130Asp) and c.521T>C (p.Val174Ala), which are associated with emerging side effects of sorafenib in HCC patients [100]. 

### 4.2. Drug-Induced Liver Injury (DILI) by Immune Checkpoint Inhibitors (ICIs)

The US FDA has approved eight ICIs for intervention, including anti-programmed death-1 [anti-PD-1] antibodies nivolumab, pembrolizumab, and cemiplimab; anti-programmed death ligand-1 [anti-PD-L1] antibodies avelumab, atezolizumab, and durvalumab; and anti-CTLA-4 antibodies ipilimumab and tremelimumab, targeting three immune checkpoints (PD-1, PD-L1, and CTLA-4) [102] (Table 3). ICIs have been used for many decades in the effective treatment of HCC. Hepatotoxicity is associated with several immune-related adverse events (irAEs) that are different from DILI and is thought to be related to the autoimmunity [102]. A previous study noted that ICI induced liver toxicity is associated with the infiltration of CD8-positive T-cells [102]. The same study reported that the incidence of ICI induced DILI is between 0.8 and 14.6% for CTLA-4 inhibitors like ipilimumab and between 2.7 and 16% for PD-1/PD-L1 inhibitors such as nivolumab [102]. Swenson and co-workers analyzed 112 patients who received durvalumab, an anti-PD-L1 antibody treatment, and observed that 19% of the patients were diagnosed with DILI using RUCAM. It is known that the risk of DILI development is directly proportional and positively correlated with the concomitant use of ICIs and chemotherapy or other ICIs [103]. Another previously published meta-analysis of 122 clinical trials reported a 0.09% mortality due to hepatitis as an irAE and 0% with an anti-CTLA-4 antibody, and 0.13% with the combination of PD-1 antibody/anti-PD-L1 antibody and an anti-CTLA-4 antibody, suggesting a non-fatal effect of combination therapies in liver damage cases [104] (Figure 7). Considering these events, using ICI-enabled treatment options could be explored for HCC cases where no other options or alternative treatments are available or where patients have high DILIs.

## 5. Conclusions

Sorafenib, used either as a monotherapy or in combination with atezolizumab and bevacizumab has remained the first choice of drug in the past decade for providing better outcomes in patients with HCC in a Asian populations. Other approaches, including cytokine-based immunotherapy, have also been explored in Asia for the treatment of HCC with minimal side effects and significant benefits. However, newer therapeutic approaches, including nanotechnology-based delivery, need to be explored further for the effective treatment of patients with HCC.
ijms-25-09286-t003_Table 3Table 3Clinical trials of immunotherapies combinations in locally advanced unresectable and metastatic HCC.
Trial Identifier

Line

Agents

Primary Endpoints

Patients

Status
**NCT03713593**First-lineLenvatinib + pembrolizumab v/s. lenvatinibPFS, OS750Ongoing**NCT03764293**First-linePD-1 antibody SHR-1210 + apatinib mesylate v/s. sorafenibPFS, OS510Ongoing**NCT03298451**First-lineDurvalumab v/s. durvalumab + tremelimumab v/s. sorafenibOS1310Active, not recruiting**NCT03412773**First-lineBGB-A317 (PD-1 antibody) v/s. sorafenibOS674Active, not recruiting**NCT03434379**First-lineAtezolizumab + bevacizumab v/s. sorafenibOS, PFS480Active, not recruiting**NCT01658878**First-lineNivolumab + cabozantinib v/s. nivolumab + ipilimumab + cabozantinibSafety, tolerability and ORR1097Active, not recruiting**NCT03347292**First-linePembrolizumab + regorafenibTEAEs, DLTs57Ongoing, recruiting**NCT03439891**First-lineNivolumab + sorafenibMTD, ORR40Ongoing, recruitingPFS, progression free survival; OS, overall survival; ORR, objective response rate; TEAEs, treatment-emergent adverse events; DLTs, dose limiting toxicities; MTD, maximum tolerated dose. (Adopted from Ref. [105] under Creative Commons Attribution-NonCommercial-4.0 International License (CC BY-NC-ND 4.0).)
Figure 7Causes and their percentage contributions to HCC in the Asian population. Data Source: Ref. [106] under Creative Commons Attribution 4.0 International License.
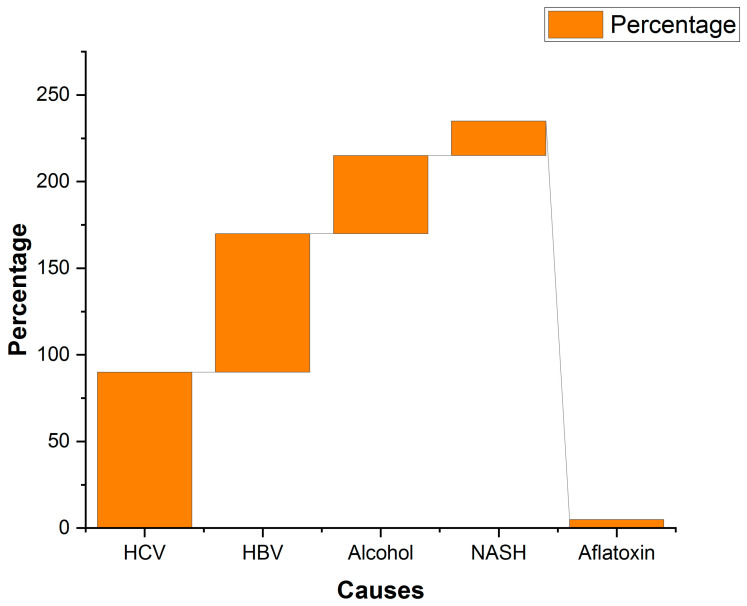



## Figures and Tables

**Figure 1 ijms-25-09286-f001:**
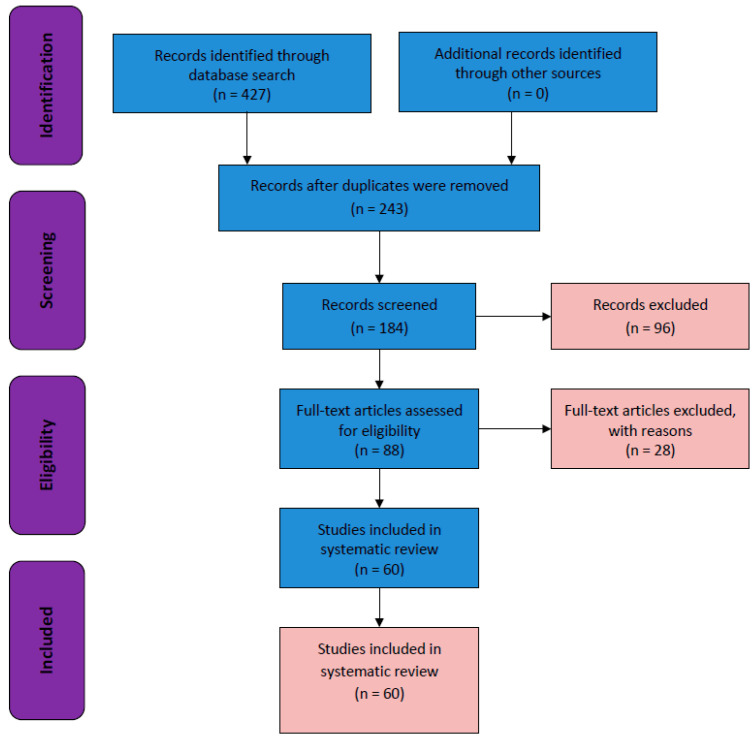
Flow chart of the Preferred Reporting Items for Systematic Reviews and Meta-Analysis.

**Figure 2 ijms-25-09286-f002:**
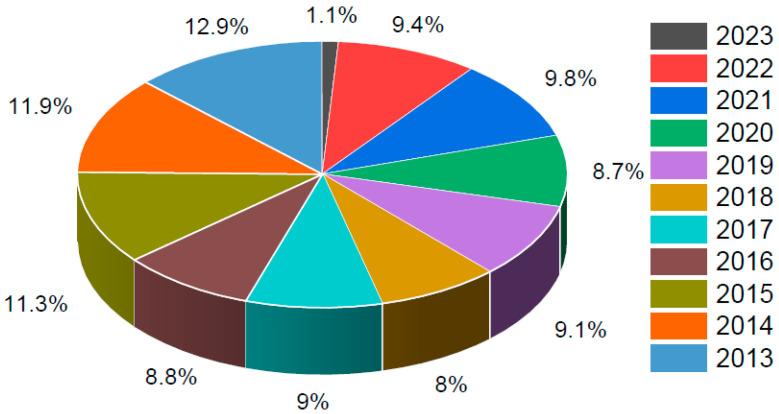
Number of publications in the last 10 years (2013 to 2023), extracted from the PubMed database, on phase I–IV clinical trials related to hepatocellular carcinoma that were conducted in South Korea.

**Figure 3 ijms-25-09286-f003:**
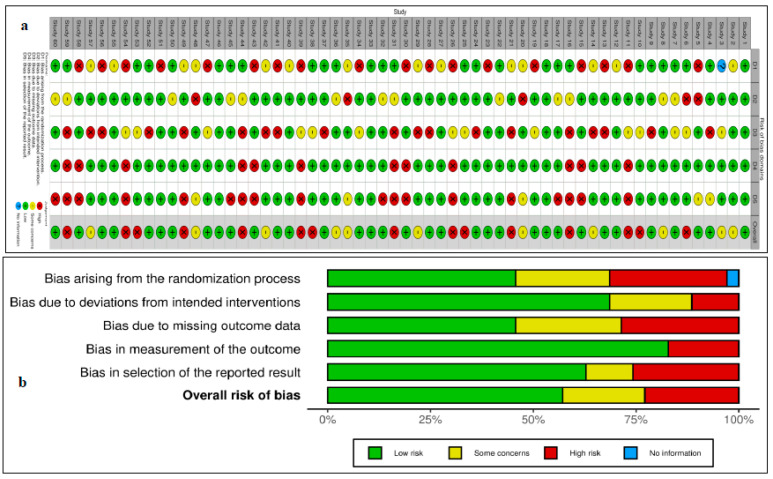
Plot demonstrating risk of bias (**a**) Traffic light plots of domain-level judgments for each individual result. (**b**) Weighted bar plots of the distribution of risk of bias judgments within each bias domain.

**Figure 4 ijms-25-09286-f004:**
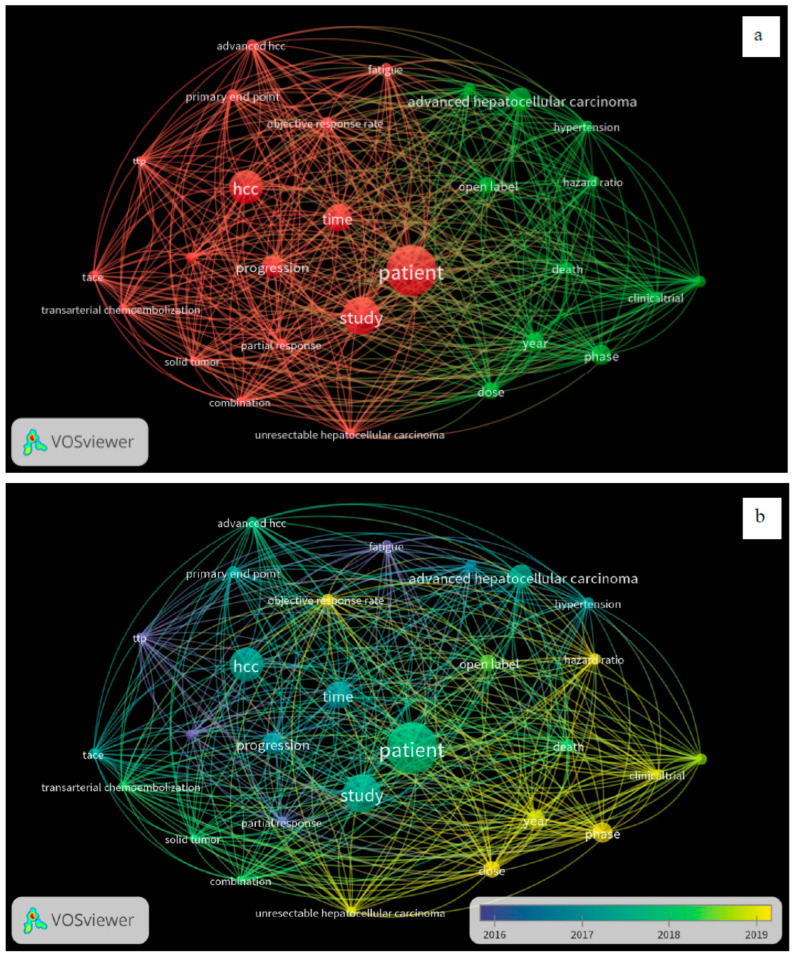
Visualization of bibliometric networks of eligible articles using VOSviewer version 1.6.16 software (n = 60): (**a**) network visualization and (**b**) overlay visualization.

**Figure 5 ijms-25-09286-f005:**
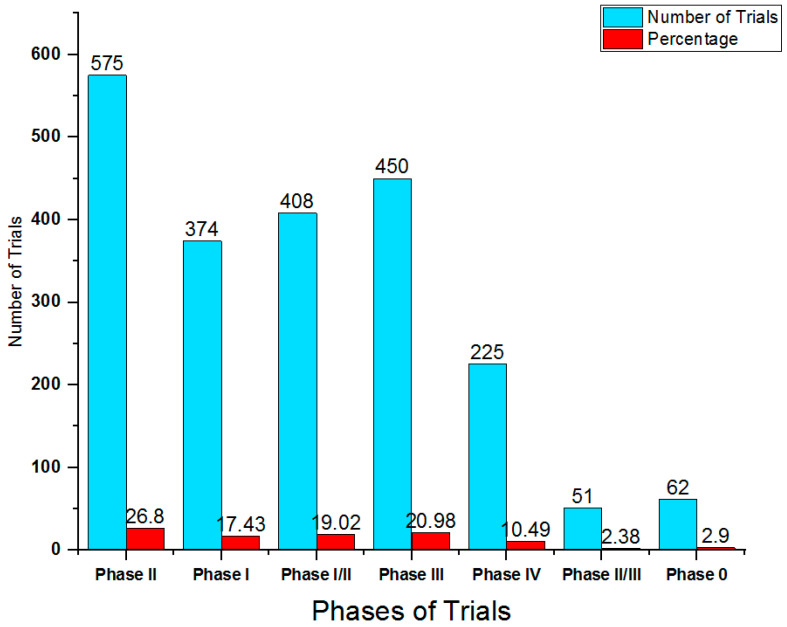
Bar representation of ongoing drug trails for HCC.

**Figure 6 ijms-25-09286-f006:**
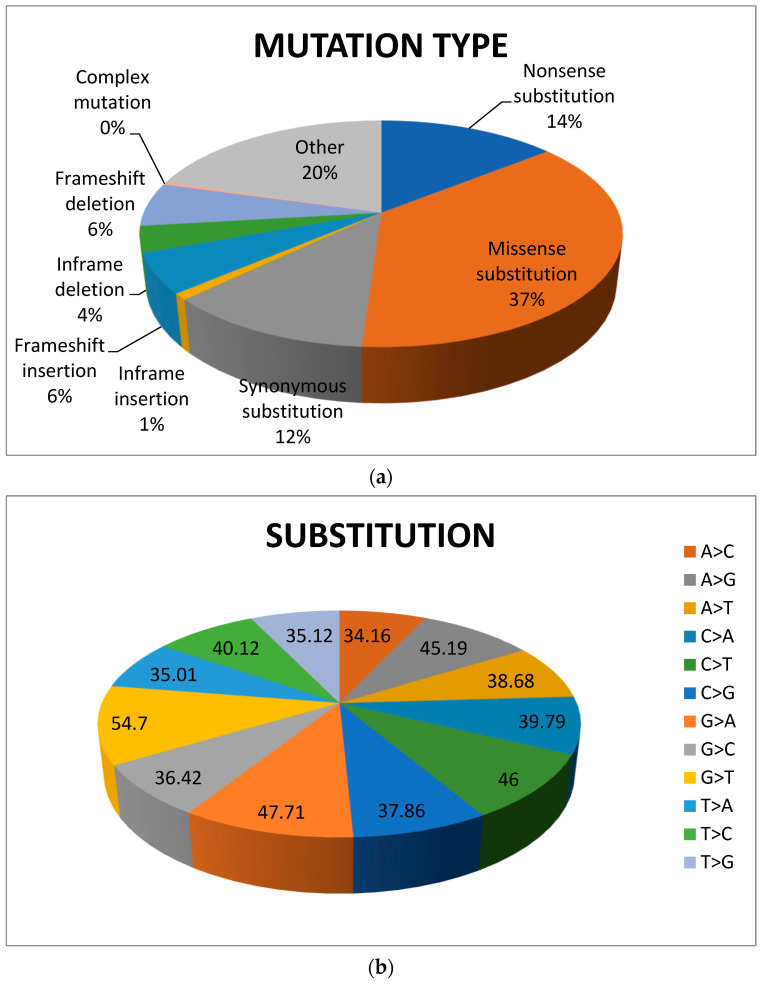
Pie chart showing (**a**) mutation and (**b**) substitution in HCC (data source: https://cancer.sanger.ac.uk/cosmic (accessed on 8 February 2024) (Ref. [101]).

**Table 1 ijms-25-09286-t001:** Eligible studies included in the systematic review showing the application in the treatment of hepatocellular carcinoma.

Author Names	Year	Drugs Used	Phase	No. of Participants (n)	Design	Dosage	References
Finn et al.	2020	Atezolizumab + Bevacizumab v/s Sorafenib	III	501	Open-label RCT	Atezolizumab = 1200 mg Bevacizumab = 15 mg Sorafenib = 400 mg	[15]
Kudo M et al.	2018	Lenvatinib v/s Sorafenib	III	468	Open-label RCT	Lenvatinib = 12 mg Sorafenib = 400 mg	[13]
Cheng AN et al.	2021	Atezolizumab + Bevacizumab v/s Sorafenib	III	501	Open-label RCT	Atezolizumab = 1200 mg Bevacizumab = 15 mg Sorafenib = 400 mg	[16]
El-Khoueiry AB et al.	2017	Nivolumab	I/II	262	Open-label, on-comparative, dose escalation and expansion trial	1–10 mg	[17]
Abou-Alfa GK et al.	2018	Cabozantinib	III	707	Double-blind, RCT	60 mg	[18]
Yau T et al.	2020	Nivolumab + ipilimumab	I/II	148	Open-label, Multicohort	Nivolumab = 3 mg Ipilimumab = 1 mg	[19]
Kelley RK et al.	2021	Tremelimumab + Durvalumab	I/II	332	Open-label RCT	Tremelimumab = 300 mg Durvalumab = 1500 mg	[20]
Lee JH et al.	2015	Autologous CIK cells	III	230	Open-label RCT	6.4 × 10^9^	[21]
Bruix J et al.	2015	Sorafenib	III	900	Randomized, double-blind, placebo-controlled trial	577 mg	[22]
Yau T et al.	2019	Nivolumab	I/II	267	Open-label RCT	3 mg	[23]
Kelley RK et al.	2022	Cabozantinib + atezolizumab v/s sorafenib	III	837	Open-label RCT	Cabozantinib = 40 mg Atezolizumab = 1200 mg Sorafenib = 400 mg	[24]
Yau T et al.	2020	Nivolumab	III	743	Open-label RCT	240 mg	[25]
Galle PR et al.	2021	Atezolizumab + Bevacizumab v/s Sorafenib	III	501	Open-label RCT	Atezolizumab = 1200 mg Bevacizumab = 15 mg Sorafenib = 400 mg	[26]
Zhu AX et al.	2019	Ramucirumab	III	197	Open-label RCT	8 mg	[27]
Lencioni R et al.	2016	Transarterial chemoembolization with doxorubicin-eluting beads (DC Bead^®^; DEB-TACE) + Sorafenib	II	307	Open-label RCT	DEB-TACE = 150 mg Sorafenib = 400 mg	[28]
Vogel A et al.	2021	Lenvatinib v/s Sorafenib	III	954	Randomized, open-label, non-inferiority	Lenvatinib = 12 mg Sorafenib = 400 mg	[29]
Finn RS et al.	2019	Pembrolizumab	III	413	Randomized, double-blind	200 mg	[30]
Lee MS et al.	2020	Atezolizumab + Bevacizumab	Ib	104	Open-label RCT	Atezolizumab = 1200 mg Bevacizumab = 15 mg	[31]
Cheon J et al.	2022	Atezolizumab + Bevacizumab	III	138	Retrospective	Atezolizumab = 1200 mg Bevacizumab = 15 mg	[32]
Park JW et al.	2019	Sorafenib	III	339	Open-label RCT	Sorafenib = 600 mg	[33]
Choi NR et al.	2022	Lenvatinib+ Sorafenib		206	Open-label RCT	Lenvatinib = 12 mg Sorafenib = 400 mg	[34]
Cheon J et al.	2020	Lenvatinib	III	67	Retrospective	Lenvatinib = 12 mg	[35]
Yoon SM et al.	2018	Sorafenib	-	99	Open-label RCT	Sorafenib = 400 mg	[36]
Hong JY et al.	2022	Pembrolizumab	II	55	Open-label RCT	200 mg	[37]
Chow PKH et al.	2018	Sorafenib	III	360	Open-label RCT	800 mg	[38]
Ryoo BY et al.	2021	Enzalutamide	II	165	Randomized, Double-blind	160 mg	[39]
Ryoo BY et al.	2021	Tepotinib v/s Sorafenib	Ib/II	117	Open-label RCT	Tepotinib = 1200 mg Sorafenib = 400 mg	[39]
Cheng AL et al.	2015	Tigatuzumab + sorafenib	II	163	Open-label RCT	Tigatuzumab = 6 mg Sorafenib = 400 mg	[40]
Cainap C et al.	2015	Linifanib v/s Sorafenib	III	1035	Open-label RCT	Linifanib = 17.5 mg Sorafenib = 400 mg	[41]
Zhu AX et al.	2015	Sorafenib + Erlotinib	III	720	Open-label RCT	Erlotinib = 150 mg Sorafenib = 400 mg	[42]
Tak WY et al.		Sorafenib + Resminostat v/s Sorafenib	I/II	179	Open-label RCT	Sorafenib + resminostat = 3 + 400 mg Sorafenib = 400 mg	[43]
Johnson PJ et al.	2013	Brivanib v/s Sorafenib	III	1150	Open-label RCT	Brivanib = 800 mg Sorafenib = 400 mg	[44]
Zhu AX et al.	2015	Ramucirumab	III	283	Randomized, double-blind	8 mg	[45]
Lim HY et al.	2014	Refametinib + Sorafenib	II	95	Open-label RCT	Refametinib = 50 mg Sorafenib = 600 mg	[46]
Chau I et al.	2017	Ramucirumab	III	565	Open-label RCT	8 mg	[47]
Qin S et al.	2020	Camrelizumab	II	220	Open-label RCT	3 mg	[48]
Qin S et al.	2021	Apatinib	III	400	Randomized, double-blind	750 mg	[49]
Llovet JM et al.	2022	Lenvatinib + Pembrolizumab	III	950	Randomized, double-blind	Lenvatinib = 12 mg Pembrolizumab = 400 mg	[50]
Ding X et al.	2021	Lenvatinib v/s Sorafenib	III	64	Open-label RCT	Lenvatinib = 12 mg Sorafenib = 400 mg	[51]
Peng Z et al.	2022	Lenvatinib	III	338	Open-label RCT	Lenvatinib = 12 mg	[52]
He M et al.	2019	Sorafenib v/s Oxaliplatin, Fluorouracil, and Leucovorin+ Sorafenib	II	818	Open-label RCT	Sorafenib = 400 mg Oxaliplatin = 85 mg Leucovorin = 400 mg Fluorouracil = 400 mg	[53]
Qin S et al.	2019	Tislelizumab v/s Sorafenib	III	640	Open-label RCT	Tislelizumab = 200 mg Sorafenib = 400 mg	[54]
Mei K et al.	2021	Camrelizumab + Apatinib	Ib/II	28	Open-label RCT	Camrelizumab = 3 mg Apatinib = 500 mg	[55]
Xia Y et al.	2022	Camrelizumab + Apatinib	II	20	Open-label RCT	Camrelizumab = 200 mg Apatinib = 250 mg	[56]
Xu J et al.	2021	Camrelizumab + Apatinib	II	120	Open-label	Camrelizumab = 200 mg Apatinib = 250 mg	[57]
Qin S et al.	2021	Donafenib v/s Sorafenib	II/III	668	Open-label RCT	Donafenib = 200 mg Sorafenib = 400 mg	[58]
Lyu N et al.	2022	Oxaliplatin+ Leucovorin +Fluorouracil v/s Sorafenib	III	262	Open-label RCT	Oxaliplatin = 130 mg Leucovorin = 200 mg Fluorouracil = 400 mg Sorafenib = 400 mg	[59]
Ren Z et al.	2021	Sintilimab + bevacizumab v/s Sorafenib	II/III	595	Open-label RCT	Sintilimab = 200 mg bevacizumab = 15 mg Sorafenib = 400 mg	[60]
Li QJ et al.	2022	Oxaliplatin + Leucovorin + Fluorouracil v/s Epirubicin + Lobaplatin	III	315	Open-label RCT	Oxaliplatin = 130 mg Leucovorin = 400 mg Fluorouracil = 400 mg Epirubicin = 50 mg Lobaplatin = 50 mg	[61]
Kang YK et al.	2015	Axitinib	II	202	Double-blind RCT	Axitinib = 5 mg	[62]
Llovet JM et al.	2013	Brivanib	III	395	Double-blind RCT	Brivanib = 800 mg	[63]
Yau TCC et al.	2017	Foretinib	I/II	32	Single-arm	Foretinib = 60 mg	[64]
Zhu AX et al.	2014	Everolimus	I	546	Open-label RCT	Everolimus = 7.5 mg	[65]
Kelley RK et al.	2020	Cabozantinib	II	331	Open-label RCT	Cabozantinib = 60 mg	[66]
Verset G et al.	2022	Pembrolizumab	II	51	Open-label RCT	Pembrolizumab = 200 mg	[67]
Abou-Alfa GK et al.	2018	Cabozantinib	III	707	Double-blind RCT	Cabozantinib = 60 mg	[18]
Tai WM et al.	2016	Selumetinib + Sorafenib	Ib	27	Open-label RCT	Selumetinib = 75 mg Sorafenib = 400 mg	[68]
Toh HC et al.	2013	Linifanib	II	44	Single-arm, open-label	Linifanib = 0.25 mg	[69]
Lim HY et al.	2018	Refametinib v/s Refametinib + Sorafenib	II	1318	Open-label RCT	Refametinib = 50 mg Sorafenib = 400 mg	[70]
Chow PK et al.	2014	Sorafenib	II	29	Open-label RCT	Sorafenib = 400 mg	[71]

RCT, randomized clinical trial.

**Table 2 ijms-25-09286-t002:** Ongoing drug trials for HCC.

Name	Number of Trials	Percentage
Phase II	575	26.80
Phase I	374	17.43
Phase I/II	408	19.02
Phase III	450	20.98
Phase IV	225	10.49
Phase II/III	51	2.38
Phase 0	62	2.90
Total Trials	2145	

Data Source: https://www.globaldata.com/data-insights/healthcare/number-of-ongoing-clinical-trials-for-drugs-involving-hepatocellular-carcinoma-by-phase-503271/ (accessed on 8 February 2024) (Ref. [90]).

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
