# Peer review of "Phase I–IV Drug Trials on Hepatocellular Carcinoma in Asian Populations: A Systematic Review of Ten Years of Studies"

_ijms, 2024, doi:10.3390/ijms25179286_

Round 1

Reviewer 1 Report

Comments and Suggestions for Authors

in this systematic review, the two authors aimed at describing the HCC therapy in Asia in the last decade.

The review is well written. Figures are appropriate. I have only minor suggestions for improving the level of the work before publication.

Although the paper is focused on the Asian population, the authors should at least mention what has happened in USA and Europe in the last decade, and haw the results of the trials have changed/improved HCC therapy. 

The Asian population has often peculiar pharmacokinetic and pharmacodynamic characteristics due to pharmacogenomic features. The author should discuss the impact of pharmacogenetics in this topic, if any.

the authors should mention and possibly summarize in a Table the novel therapies which are in advanced phase of clinical development. 

Comments on the Quality of English Language

English is fine, but some typos are present. Just critically check the text and revise 

Author Response

Comments 1: [Although the paper is focused on the Asian population, the authors should at least mention what has happened in USA and Europe in the last decade, and haw the results of the trials have changed/improved HCC therapy.]

Response 1: [I included the suggested part in the review ] Thank you for pointing this out. I/We agree with this comment. Therefore, I/we have.[The revised paragraph was added to page 13]”

Comments 2: [The Asian population has often peculiar pharmacokinetic and pharmacodynamic characteristics due to pharmacogenomic features. The author should discuss the impact of pharmacogenetics in this topic, if any.

.]

Comments 3: [the authors should mention and possibly summarize in a Table the novel therapies which are in advanced phase of clinical development]

Response 2: Agree. I/We have, accordingly, done/revised/changed/modified…..to emphasize this point. We added table 3]”

Reviewer 2 Report

Comments and Suggestions for Authors

Interesting review on a hot topic. I have some issues to be discussed in the text.

Major points:

1. Please include in the text the percentages of causes leading to HCC in Asia.

2. Likely HCC often occurred in connection with HBV or HCV infection. Did these patients receive antiviral therapy before?

3. Stratify efficacy of HCC therpy with respect to etiology

4. Please mention ADRs  specifically with respect to DILI and whether this was evaluated by RUCAM.

5. Discuss the high risk of ICIs for DILI.

Comments on the Quality of English Language

English needs improvements. 

Author Response

Comments 1: [Please include in the text the percentages of causes leading to HCC in Asia..]

Response 1: [Type your response here and mark your revisions in red] Thank you for pointing this out. I/We agree with this comment. Therefore, I/we have.[We have added Figure 7.]

“[updated text in the manuscript if necessary]”

Comments 2: [Likely HCC often occurred in connection with HBV or HCV infection. Did these patients receive antiviral therapy before?

Response 2: Agree. I/We have, accordingly, done/revised/changed/modified…..to emphasize this point. No Patients did not received any antiviral therapy before ]”

Comments 3: [Stratify efficacy of HCC therapy with respect to etiology?]

Response 3: Agree. I/We have, accordingly, done/revised/changed/modified…..to emphasize this point. Added on Page 15 and 16 ]”

Comments 4: [Please mention ADRs  specifically with respect to DILI and whether this was evaluated by RUCAM]

Response 4: Agree. I/We have, accordingly, done/revised/changed/modified…..to emphasize this point. Added on Page 15 and 16 ]”

Comments : [Discuss the high risk of ICIs for DILI.]

Response 5: Agree. I/We have, accordingly, done/revised/changed/modified…..to emphasize this point. Added on Page 15 and 16 ]”

Round 2

Reviewer 2 Report

Comments and Suggestions for Authors

Perfect revision

Comments on the Quality of English Language

Minor English  modification is needed